# Subtype-specific characterization of breast cancer invasion using a microfluidic tumor platform

Hye-ran Moon[1], Natalia Ospina-Muñoz[1,2], Victoria Noe-Kim[1], Yi Yang[3], Bennett D. Elzey[4,5], Stephen F. Konieczny[3,5], Bumsoo Han[1,5,6]*

1 School of Mechanical Engineering, Purdue University, West Lafayette, IN, United States of America, 2 Cellular and Molecular Physiology Group, School of Medicine, Universidad Nacional de Colombia, Bogotá D.C, Colombia, 3 Department of Biological Science, Purdue University, West Lafayette, IN, United States of America, 4 Department of Comparative Pathobiology, Purdue University, West Lafayette, IN, United States of America, 5 Purdue Center for Cancer Research, Purdue University, West Lafayette, IN, United States of America, 6 Weldon School of Biomedical Engineering, Purdue University, West Lafayette, IN, United States of America

* bumsoo@purdue.edu

**Data Availability Statement:** All relevant data are within the manuscript and its Supporting Information files.

## Abstract

Understanding progression of breast cancers to invasive ductal carcinoma (IDC) can significantly improve breast cancer treatments. However, it is still difficult to identify genetic signatures and the role of tumor microenvironment to distinguish pathological stages of pre-invasive lesion and IDC. Presence of multiple subtypes of breast cancers makes the assessment more challenging. In this study, an *in-vitro* microfluidic assay was developed to quantitatively assess the subtype-specific invasion potential of breast cancers. The developed assay is a microfluidic platform in which a ductal structure of epithelial cancer cells is surrounded with a three-dimensional (3D) collagen matrix. In the developed platform, two triple negative cancer subtypes (MDA-MB-231 and SUM-159PT) invaded into the surrounding matrix but the luminal A subtype, MCF-7, did not. Among invasive subtypes, SUM-159PT cells showed significantly higher invasion and degradation of the surrounding matrix than MDA-MB-231. Interestingly, the cells cultured on the platform expressed higher levels of CD24 than in their conventional 2D cultures. This microfluidic platform may be a useful tool to characterize and predict invasive potential of breast cancer subtypes or patient-derived cells.

## Introduction

Breast cancers often start from ductal carcinoma *in situ* (DCIS), in which abnormal epithelial cells are present in the mammary ducts of breast tissues. [1, 2] This pre-cancerous lesion progresses to invasive ductal carcinoma (IDC), which will need major treatments including chemotherapy, radiation, and surgical therapies. [3–5] Understanding and characterization of the transition of DCIS into IDC can significantly improve the treatment and management of breast cancers. Extensive studies have been performed to identify both genetic and

**Funding:** This work was partially supported by NIH HHSN261201400021C, a CTR Award from Indiana CTSI funded in part by UL1 TR000006 from NIH, a Phase I grant from the Purdue University Center for Cancer Research, and a grant from Walther Cancer Foundation. The confocal microscopy was by a Shared Resource Award from the Purdue University Center for Cancer Research, NIH grant P30 CA023168. Natalia Ospina-Muñoz was partially supported by COLCIENCIAS. National PhD, Call 647 Year 2015.

**Competing interests:** The authors have declared that no competing interests exist.

environmental drivers and markers of the progression from DCIS to IDC, as reviewed else-where. [1, 2, 6] However, key drivers of this disease's progression along with both pathological and molecular markers remain elusive. Moreover, key markers vary depends on breast cancer subtypes, which are poorly understood.

Invasion of malignant epithelial cells from a mammary duct is a complex process, which consists of cell growth, epithelial-mesenchymal-transition (EMT), cell invasion, and migra-tion. During this process, various growth factors and enzymes are thought to be involved. [7, 8] It has been reported that notable remodeling of the surrounding extracellular matrix (ECM) accompanies this process. [9–11] Although changes in gene expressions of various cell types in DCIS and IDC have been studied, [10, 11] many studies have reported very similar gene expression levels in both a pre-invasive lesion and IDC. DCIS shows similar genetic aberra-tions and intralesional heterogeneity to synchronous IDC. [12–15] This makes it challenging to identify relevant genetic signatures that help distinguish pathological stages between DCIS and IDC.

Furthermore, invasive and metastatic behaviors are largely varied depending on the cancer subtypes. Breast cancers have been classified into several subtypes based on histological con-texts, genetic profiles, or molecular aberrations. [16–19] Within a histologically identical tumor type, particularly IDC, clinical outcomes have shown drastically different. [16, 18] For this reason, several molecular subtypes of breast cancer defined by gene expression profiles including luminal A, luminal B, HER-2 enriched, basal-like, and triple-negative breast cancer (TNBC) type have aroused to improve the therapeutic strategies. Specifically, TNBC, which lacks expression of hormone epidermal growth factor receptor 2 (HER-2), ER and PR, has drawn enormous attention due to its aggressiveness. [18, 20] Even within the TNBC subgroup, it has shown heterogeneity in invasive behaviors and clinical outcomes. [18, 21] However, quantitative comparison of subtype-specific invasion process has been barely studied due to lack of the capable models. It is highly desired to develop subtype-specific invasion models that can provide patient-specific or subtype-specific treatment planning.

One of the most crucial gaps in elucidating mechanisms of the complex invasion process is the lack of a relevant tumor model capable of recapitulating series of events that occur during the local invasion of cancer cells. Conventional 2D cell cultures are easy to use but limited to representing cellular activity during this progression due to the lack of relevant environmental cues present in the tumor environment. In-vivo models, such as xenografts, [8, 9] provide a better representation of human breast cancers, but it is difficult to control environmental con-ditions and recapitulate anatomical details of the IDC microenvironment. Thus, new tumor models are highly desired to be capable of recapitulating the interactions of epithelial cells and ECM while maintaining the controllability of environmental cues. In this context, microfluidic tumor models offer significant advantages in enabling recapitulation of the tumor microenvi-ronment through precise control of physiological cues. [22, 23] Several microfluidic models have been recently used to characterize subtype-specific drug response of breast cancers, [24–26] and subtype-specific metastasis to bone. [27] Regarding DCIS, several microfluidic models have been proposed to study the effects of fibroblast on the invasion of cancer and non-cancer-ous epithelial cells of breast tissues. [28–30] The platform in [29] has microchannels where fibroblasts are cultured in collagen to induce chemical cues on the epithelium of the duct. In addition to the DCIS, an *in vitro* microfluidic tumor model for the later stage IDC has devel-oped to investigate invasion characteristics of the breast cancer cells. Despite the advances in developing *in vitro* tumor models, a lack of a reliable model to explore the subtype-specific fea-tures of the breast cancer invasion is available.

In this study, we developed an *in vitro* model of IDC and used to quantify subtype-specific invasion characteristics of three different human breast cancer cell lines–MCF-7, MDA-MB-

231, and SUM-159PT. This IDC-on-chip model is a microfluidic platform in which a breast cancer epithelial cell duct is surrounded with three-dimensional (3D) perfused collagen matrix. This model is capable of closely mimicking pathobiology of cancer cells, including growth, epithelial-mesenchymal transition (EMT), local invasion, and associated signaling pathways. Cancer cell lines that are utilized in this study are critical molecular subtypes of breast cancers. Since the developed model intends to mimic IDC which is a later stage of the disease after DCIS, the three subtypes of breast cancer cell lines were selected accordingly. MCF-7 cell line is the Luminal A subtype expressing estrogen receptor (ER) and progesterone receptor (PR). Both MDA-MB-231 and SUM-159PT are TNBCs. [31, 32] These two cell lines are further characterized as mesenchymal stem-like TNBC subtypes. [33, 34] Subtype-specific local invasion characteristics of these cells into the surrounding matrix were assessed quantitatively.

## Materials and methods

### Cells and reagents

Three types of human breast cancer cell lines (MCF-7, MDA-MB-231, and SUM-159PT) were used. MCF-7 cells were maintained in a culture medium (DMEM/F12, Invitrogen) supplemented with 5% fetal bovine serum (FBS), 2 mM L-glutamine, 100 μg/mL penicillin/ streptomycin. The culture medium for MDA-MB-231 cells was supplemented with 10% FBS. SUM-159PT cells, obtained from Asterand (Detroit, MI), were cultured in a medium (Ham's F-12, Invitrogen) supplemented with 5% FBS, 10mM HEPES, 5μg/ml insulin, and 1μg/ml hydrocortisone (Sigma-Aldrich, St. Louis, MO). All cells were cultured in 75 cm$^2$ T-flask at 37˚C and 5% $CO_2$. Cells were harvested for further experiments using 0.05% trypsin and 0.53 mM EDTA when the cells reached 70–80% confluence.

### Microfluidic invasion assay

The IDC-on-chip was developed to replicate a mammary duct with an intact epithelial cell lining that is supported and surrounded by a collagen matrix as shown in Fig 1A. The microchannel (width and height = 900 μm, length = 10 mm) was fabricated with poly-dimethylsiloxane (PDMS) using a micromachined mold. The channel was initially filled with type I collagen. Prior to gelation, a lumen structure, to represent the mammary duct, was generated along the microfluidic channel using viscous fingering as described previously. [29, 35] Briefly, a solution of 6.0 mg/mL Type I collagen (Corning, NY) was then prepared. Approximately 20 μL of collagen solution was added to the inlet port of the device so that the channel was filled with collagen. Then, a single droplet of 20 μL of culture medium was placed at the inlet causing a duct structure to form through the collagen via viscous fingering. The devices were then incubated at 37˚C for 15–20 minutes for gelation of the collagen solution to occur. Once the ductal structure was formed, 10 μL of cancer cell suspension was added to the inlet of the device to seed cells on the lumen, which was subsequently incubated at 37˚C and 5% $CO_2$. Culture medium was changed every 24 hours until the conclusion of the experiment.

### Invasion characteristics and confocal microscopy

The extent of local invasion of cancer cells was quantified via image analysis using ImageJ (NIH, Bethesda, MD). The brightfield images obtained were further analyzed to quantify local invasions. The local invasion was quantified by counting sprouting from the lumen boundary into the adjacent collagen matrix. The number and length of sprouting from the lumen were measured. The number of sprouting sites was normalized across all samples, yielding a "local

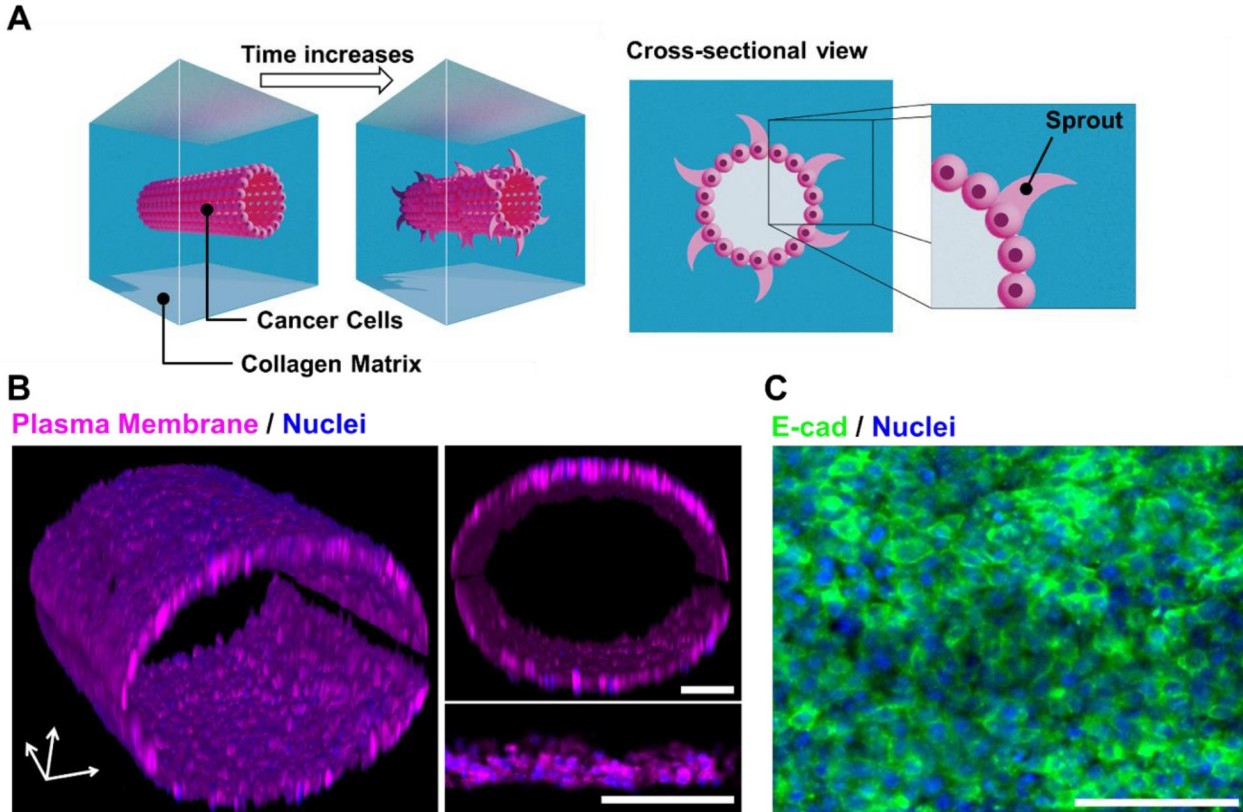

**Fig 1. Invasive ductal carcinoma (IDC) on chip.** (A) Microanatomical replication of IDC-on-chip. Epithelial cancer cells at the lining of the lumen structure formed through collagen gel. (B) 3D reconstruction of MCF-7 lumen structure in IDC-on-chip platform and cross-sectional images. Magenta stains are plasma membrane and blue are nuclei. (C) E-cadherin (E-cad, green) immunostaining of MCF-7 duct with nuclei (blue). Scale bars indicate 100μm.

invasion score" between 0 and 1. Mean sprout length was determined as the straight-line distance between the base and tip of the local invasion site.

Prior to terminal confocal microscopy, cells in IDC-on-chip were labeled for cell nuclei (Hoechst 33342, Sigma Aldrich, MI) and plasma membrane (Deep Red Cell Mask, Life Technologies, CA). The stained samples were fixed with 10% formaldehyde. Z-stacked confocal images were acquired with 2.5–4 μm intervals using a confocal microscope (A1R-MP, Nikon, Japan). At each focal plane, fluorescence illuminations were obtained. To highlight the invasive configuration of the cells, fields of view of confocal fluorescence were reconstructed as 3D.

## Immunofluorescence microscopy

For immunofluorescence microscopy, the IDC-on-chip samples were pre-cultured for three days. One day prior to immunostaining, the samples were fixed by placing a few droplets of 10% buffered formalin (MACRON, Cat:5016–08, PA) at the inlet until the culture medium was completely replaced. After fixation, samples were washed three times with 1X PBS (Life Technologies, CA) and stored with PBS reservoirs at both inlet and outlet ports to prevent dehydration, then placed them in the refrigerator at 4˚C. Once samples were ready for immunostaining, the samples were permeabilized with (Triton-X) in 1X PBS. Then the samples were blocked with a 10 μL droplet of 1% BSA and Avidin (Streptavidin) solution by adding it to the inlet of samples and allowed to incubate for 1 hour at room temperature. The sample was then

washed three times with 1X PBS. For the next step, the primary antibodies, E-cadherin (Abcam, MA) were added to M.O.M. diluent (Vector Laboratory BMK2202) and 4 drops of Biotin solution per 1 mL of diluent at a 1:100 dilution. The solution was applied to the inlet making sure it perfused through the channel then allowed to incubate for 1 hour at room temperature. After washing the channel three times with 1X PBS, a few droplets of the secondary antibody mixture and a biotin-conjugated secondary (goat) antibody (Vector Laboratory BA9200) (1:200) in M.O.M. diluent were introduced to the samples and allowed to incubate for 30 minutes at room temperature. Finally, samples were washed with 1X PBS three times, and then the stained with DAPI (1:200 in M.O.M. diluent) (Invitrogen D1306) and tertiary antibodies of Alexa Fluor 488, then incubated for 20 minutes at room temperature. Samples were washed with PBS three times before imaging.

## Histology processing and immunohistochemistry

After culturing for 72 hours, IDC-on-chip samples were incubated with media containing 10μM BrdU (5-bromo-2'-deoxyuridine) for 1.5 hours and subsequently subjected to fixation in 10% neutral buffered formalin at 4°C overnight. The samples were then filled 15% native page acrylamide gel for maintaining duct structure. The scalpel was used to cut open the PDMS chips and expose the luminal collagen tissue. The samples were then processed, embedded in paraffin, sectioned using standard histological techniques. Sections were deparaffinized and retrieved using the 2100-retriever (Electron Microscopy Sciences, Hatfield, PA) with antigen unmasking solution (Vector Laboratories, Burlingame, CA). After processing, the samples were further stained for BrdU, Ki67, and N-cadherin.

## Flow cytometry analysis

Cell surface analysis was performed to compare the phenotypic changes induced by culturing on the IDC-on-chip platform against that of conventional 2D monolayer culture. Briefly, after 3-day culture, the cells attached to the collagen matrix were retrieved from the IDC-on-chip platform. Then, the samples were treated with 0.5 mg/ml collagenase (Liberase TM, Roche, Indianapolis, IN) and 50 μg DNase I (Sigma, St. Loius, MO) for 15 min at 37°C with shaking to isolate cancer cells. Collagenase was neutralized with cold culture medium containing 10% calf serum and single cells were isolated. Cells were stained with fluorescently labeled anti-CD24, and anti-CD44 (Biolegend, CA) antibodies for 30 min at 4°C, washed twice and fixed prior to analysis using flow cytometry (LSR II cytometer, Beckton Dickinson, CA).

## Statistical analysis

Relevant study groups in each experiment were compared using a two-sample $t$-test. A minimum of three samples was analyzed for each group. The minimum level for statistical significance was $p < 0.05$. Data is reported in the form of mean ± standard error.

## Results

### Development of a microfluidic platform to characterize invasion behaviors of three different breast cancer subtypes

The mammary gland is composed of epithelial ductal trees surrounded by a complex stroma. The ducts consist of a layer of epithelial cells in the lining covered by myoepithelial cells and basement membrane (BM) [36]. In the mammary gland, the BM is a specialized laminin-rich form of the extracellular matrix. However, abnormal microenvironments are characterized by the presence of collagen I [36]. The IDC-on-chip was designed to replicate the mammary duct

with an intact epithelial cell lining that is supported and surrounded by collagen I matrix as shown in Fig 1A. The IDC-on-chip was used to evaluate the invasion capacity of three different breast cancer cell lines MCF-7, MDA-MB-231 and SUM-159PT. The ductal structure of IDC-on-chip was characterized by reconstructing the confocal fluorescence image of MCF-7 in the IDC-on-chip. In the sample, MCF-7 cells are attached on the tubular surface of the collagen matrix. Stained nuclei and cell plasma membrane clearly demonstrate a ductal structure developed in the IDC-on-chip as shown in Fig 1B. It is noted that the lumen is a cylindrical, hollow structure. MCF-7 cells showed a tightly packed epithelium with mono or double cell layers. The average diameter of the lumen was approximately 500 μm.

To confirm intact adhesion between cells, the expression of E-cadherin was evaluated. The cells in the duct showed a positive expression of E-cadherin. Indeed, the significant signal was observed at the membrane junction between the adjacent cells as shown in Fig 1C, indicating cell-cell adhesion. The ductal structure formed in IDC-on-chip has a micro-anatomical resemblance to human DCIS, which is defined as a neoplastic lesion with proliferative tumoral cells confined to the duct of the breast without invasion of the basement membrane. [37]

## MCF-7, MDA-MB-231, and SUM-159PT showed differences in the invasion capacity in the IDC-on-chip

Here, we evaluated invasion characteristics of breast cancer cells by using the 3D *in vitro* structure of IDC-on-chip. The lumen of the duct was lined with MCF-7, MDA-MB-231, and SUM-159PT cells, separately. The platform enabled the observation of the invasion capacity of the cells from the duct into the collagen I (stroma) which resembles what happens *in vivo*. Since the breast cancer cells invade through the ECM in response to its stiffness, cell sprouts indicating invasion in the IDC-on-chip headed outward of the duct. The stiffness of the collagen matrix at 6 mg/mL has been measured at $0.75 \pm 0.12$ kPa. [38] Representative time-lapsed micrographs are shown in Fig 2A. MCF-7 cells showed an increase in the thickness of the cell ductal layer across 6 days of culture, suggesting the growth of the tumoral cells inside the duct. Additionally, the collagen matrix was preserved across the duct over 6 days while no evident sprouts were formed as in Fig 2A (i, ii, iii). For MDA-MB-231, sprouts were observed after 3 days of culture (Fig 2A, v) and the number of sprouts increased over time until day 6 (Fig 2A, vi). Interestingly, SUM-159PT showed sprouts from day 1 (Fig 2A, vii), and the number of sprouts massively increased over time. The sprouts were quantitatively represented as local invasion score and local invasion length, which were defined as a frequency of the sprouting sites and a mean length of the sprouts. The quantitative evaluation of the sprouts with the frequency and length illustrates the invasion capacity of each cell line meticulously. On day 1 and day 3, SUM-159PT had a higher invasion score than MDA-MB-231 cells. (Fig 2B) The length measurement of the sprouts was similar between MDA-MB-231 and SUM-159PT, however, the morphology of the sprouts from SUM-159PT appeared different from the sprouts formed by MDA-MB-231. For SUM-159PT, sprouts had ramifications differently from MDA-MB-231 in which the sprouts looked like an extension of the cytoplasm without ramifications. These results showed differences in the invasion capacity between the different cell lines despite both MDA-MB-231 and SUM-159PT are classified as TNBC.

## Different invasion patterns between SUM-159PT and MDA-MB-231

We observed differences in sprout features and the invasion capacity between MDA-MB-231 and SUM-159PT. Specifically, SUM-159PT is highly aggressive compared to MDA-MB-231. To closely look into the invasion capacity of the cell lines, we investigated the confocal fluorescence microscopy images reconstructed in 3D for SUM-159PT (Fig 3A) and MDA-MB-231

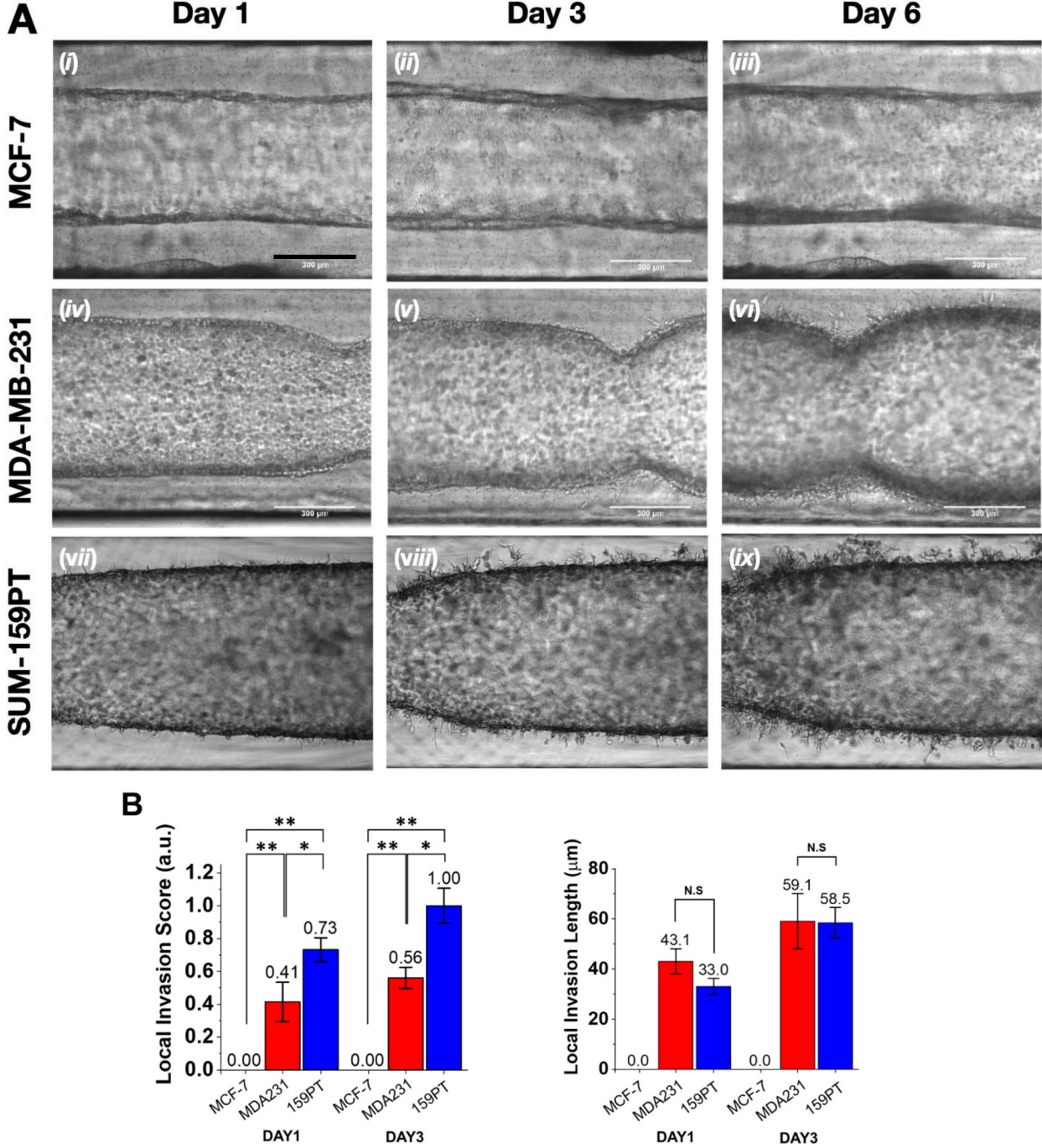

**Fig 2. Invasion assay of three subtype-specific breast cancer cell lines in IDC-on-chip.** (A) Time-lapse micrographs of MCF-7, MDA-MB-231 and SUM-159PT ducts in IDC-on-chip at day 1, 3 and 6. Scale bars indicate 300 μm. (B) Quantification of local invasion on DAY1 and DAY3. (C) Quantification of local invasion length. MCF-7; MDA231, MDA-MB-231; 159PT, SUM-159PT; Bar; mean ± standard error. N.S; p>0.05, *; p < 0.05, and **; p < 0.01 (Student t-test).

(S1 File). The duct of SUM-159PT showed protrusions that invaded the collagen as a cluster of cells aligned one after another by forming branches. (Fig 3B). A video clip of the local invasion of SUM-159PT as in S1 Video also clearly showed the invasions of clustered cells toward the collagen matrix, suggesting a collective migration. The collective migration is characterized by

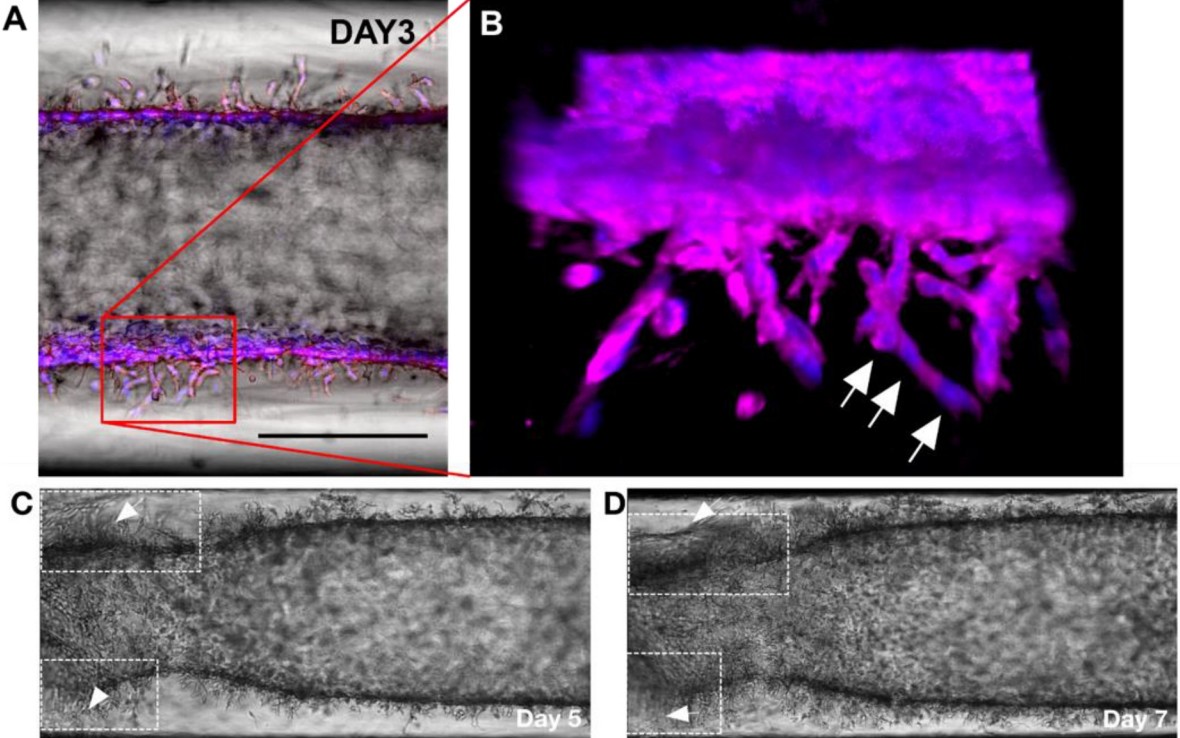

**Fig 3. Invasion characteristics of SUM-159PT in IDC-on-chip.** (A) Bright field image of SUM-159PT cells in IDC-on-chip at day 3. Scale bar is 300 μm. (B) Confocal micrograph of SUM-159PT at day 3, cells showed characteristics of multicellular collective migration in the protrusion. Cell nuclei along a protrusion are noted with white arrows. Nuclei (blue) and plasma membrane (magenta). (C and D) The duct of SUM-159PT showed regions of matrix collapse, which are thought to be associated with matrix degradation (noted with arrows), it was observed from day 5 to day 7.

the movement of cell clusters. Indeed, such movement of cells is shown *in vivo* as a disorganized strand-like mass specifically in the metastatic stage. [39, 40] Furthermore, we noticed that some regions of the collagen collapse from day 5 to day 7, which could be associated with matrix degradation. As one of the key processes supporting the cell invasion, the observation of cell interaction with the matrix degradation supports the considerable potential for SUM-159PT to invade the surrounding matrix. (Fig 3C and 3D) On the other hand, the duct of MDA-MB-231 in IDC-on-chip showed elongated membrane protrusions from individual cells which was confirmed by the presence of a single nucleus for the protrusion as shown in S1 File. The invasive capacity of SUM-159PT, projected in the collective migration and matrix degradation which not shown in MDA-MB-231, implies the invasive and metastatic behaviors can be significantly varied within the TNBC subtypes.

## Analysis of cell proliferation and expression of N-cadherin of SUM-159PT cells in IDC-on-chip

One important aspect of the duct in the mammary gland is that it is compose by a monolayer of epithelial cells. Focused on this aspect, we evaluated by Haematoxylin and Eosin staining the characteristics of the duct, we confirm that the duct is formed by a monolayer of cells Fig 4A. Furthermore, we evaluated proliferation Fig 4B and N-cadherin expression (an important protein for migration and cell-cell adhesion) Fig 4C. [41] Migration and proliferation are mutually exclusive biological process. [42] For cell proliferation we used the BrdU marker

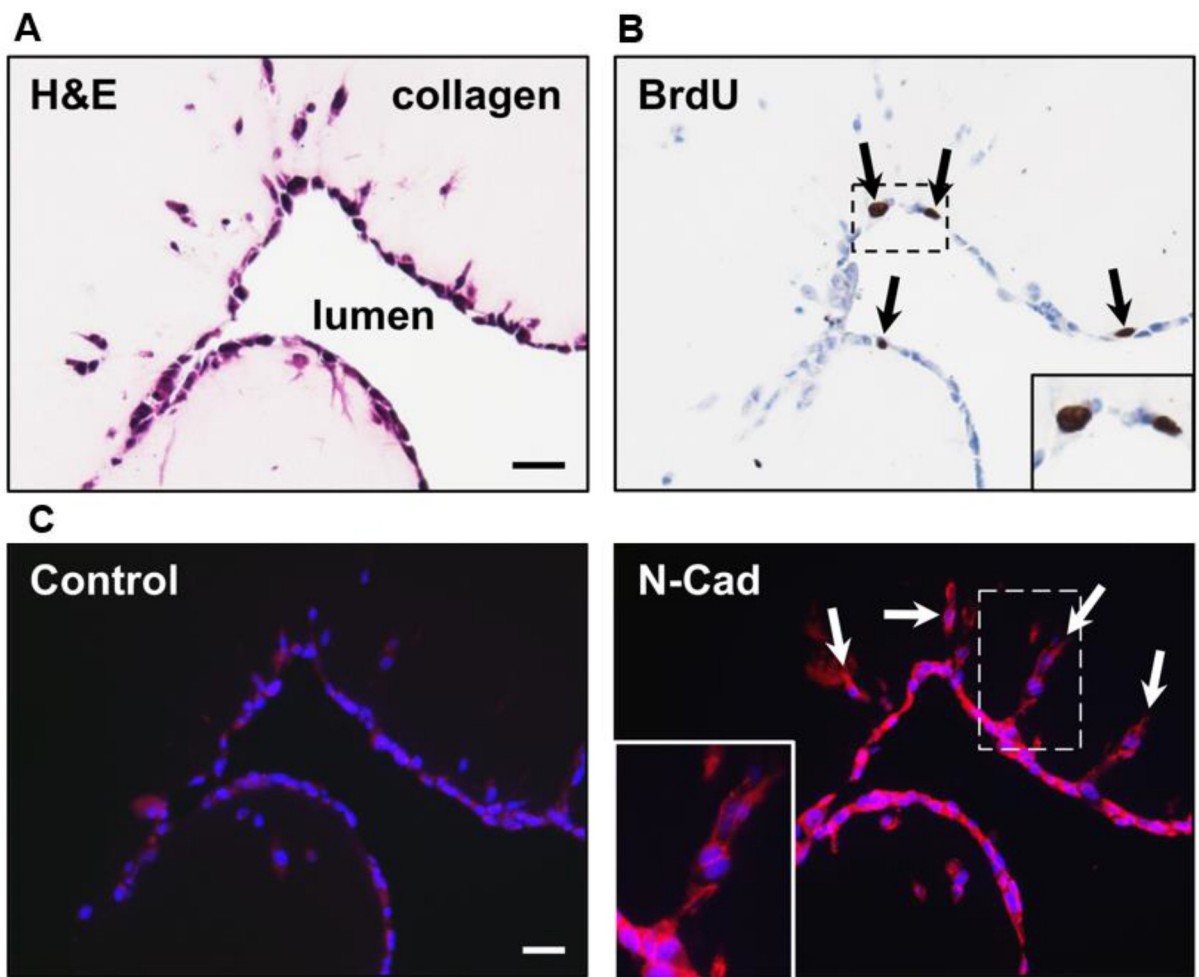

**Fig 4. Histology and Immunohistology of SUM-159PT cells in IDC-on-chip.** (A) Structure and cell morphology at the duct in IDC-on-chip. Histological staining of the cells in the duct was performed with Haematoxylin and Eosin (H&E) at day 3 of culture. (B) Evaluation of cell proliferation using BrdU assay. (C) Expression of N-cadherin at the protrusions. Nuclei (blue) and N-cadherin (red). N-cadherin is a cell-cell adhesion marker important for migratory and invasive behavior of tumoral cells.

(BrdU show cells in S phase of cell cycle). We found that some cells at the duct in the IDC-on-chip were proliferating (showed with black arrows) and others were in a non-proliferative state. Cells at the protrusions showed a negative staining for BrdU, which suggest that the migrating cells were in a non-proliferative state. There is a switch between these two cellular fates. [42] Analysis of N-cadherin expression showed that cells at the protrusion express N-cadherin, this support the idea of a collective migration behavior by the SUM-159PT cell line.

## Differences in the expression between CD44 and CD24 in 2D culture and IDC-on-chip

The use of a 3D platform like the IDC-on chip for the culture of breast cancer cells are more useful because it resembles some important physiological characteristics and allow the investigation of complex interactions between tumoral cells and stroma. [43] We made the comparison between 2D culture and the IDC-on-chip (Fig 5). Here, we noticed some differences in the expression of CD44 and CD24 between MDA-MB-231 and SUM-159PT. MDA-MB-231 cells

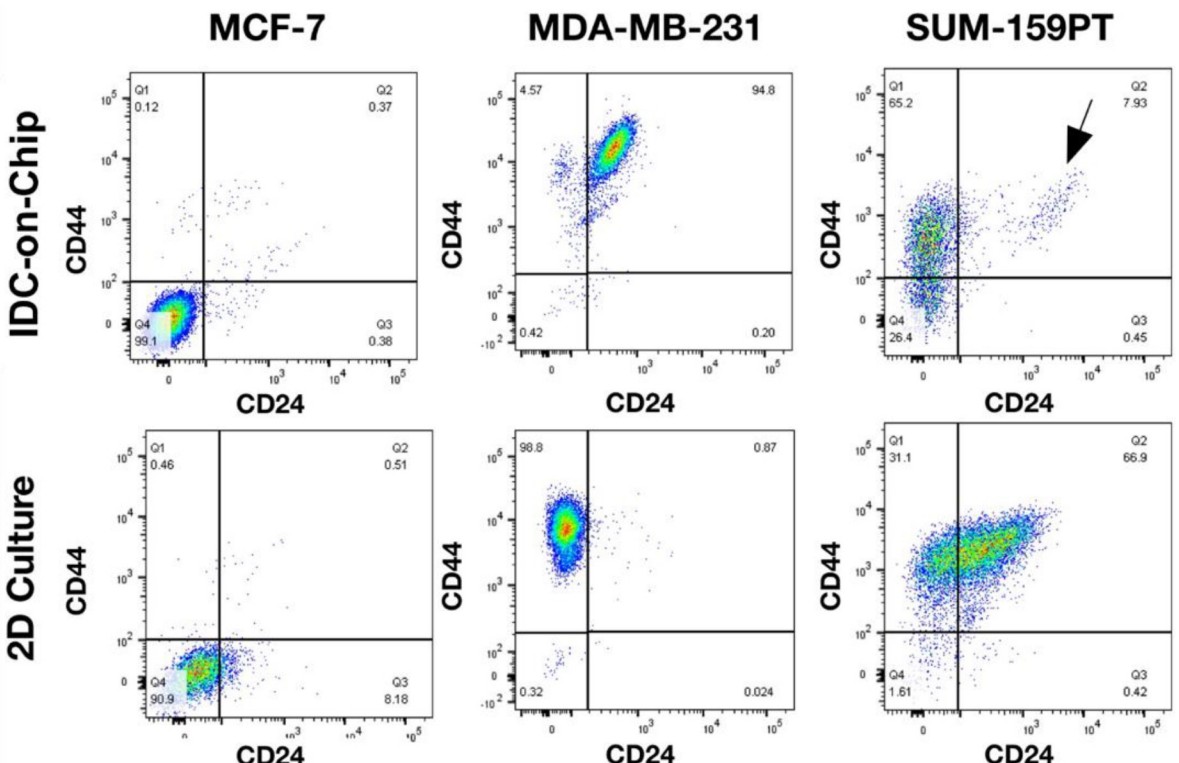

**Fig 5. Comparison of the expression of cell surface markers CD44, and CD24 between 2D culture and IDC-on-chip for the three subtype-specific breast cancer cell lines.**

culture on IDC-on-chip showed an increase in the expression of CD24, compared to 2D culture. MDA-MB-231 cells on IDC-on-chip were predominantly CD44+/CD24+. In contrast, SUM-159PT cells culture on IDC-on-chip showed an increase in the expression of CD44, compared with 2D culture. SUM-159PT cells on IDC-on-chip were predominantly CD44 +/CD24-. CD44+/CD24- expression is associated with cancer stem like properties, chemoresistance and more invasive capacity. [44–46] Which is in accordance with the results of the evaluation on the invasion capacity of SUM-159PT. However, despite the phenotype of MDA-MB-231 in the IDC-on-chip (CD44+/CD24+), this cell line also showed invasion capacity in the IDC-on-chip. Similar, the analysis of the expression of CD44 and CD24 MCF-7 showed no differences between 2D and IDC-on-chip. These results suggest that there are many factors important for the invasion capacity of the cells not only the expression of CD44 and CD24 and that the IDC-on-chip is a good platform to evaluate invasion capacity of tumoral cells.

## Discussion

The present model is capable of recapitulating cancer invasion as key pathophysiological processes as well as predicting the invasive potential of subtypes of breast cancer cells. Among three cell lines studied, the MCF-7 cell line is considered non-invasive and non-tumorigenic *in vivo* unless supplemented with estrogen. [16] Thus, the IDC-on-chip platform well predicts the MCF-7 cell line's non-invasive characteristics. On the contrary, TNBCs are thought to be more aggressive and invasive than other breast cancer subtypes. Since TNBCs are a highly diverse group of cancers, several molecular subtypes are identified within TNBCs. [16, 20, 33,

34] Both MDA-MB-231 and SUM-159PT are mesenchymal stem-like subtypes, which strongly express genes for EMT, cell motility (Rho pathway), and cellular differentiation. [16, 20] In the IDC-on-chip platform, both TNBC cell lines show distinct invasive characteristics, suggesting a capacity of the present model to characterize the invasive potential of breast cancer cells in quantitative manners. Remarkably, higher invasion capacity was observed in SUM-159PT compared to MDA-MB-231, showing collective migration and collagen matrix degradation. Although further investigation is needed to identify the differences in their molecular mechanisms for invasion, the quantitative comparison for the subtypes demonstrates the capacity of the IDC-on-chip model to develop subtype-specific or patient-specific therapeutic strategies further.

The processes, which are mimicked in the present model, include EMT, remodeling of the surrounding matrix, collective migration of cancer cells. Considering that there are currently no apparent biomarkers to predict the progression from DCIS to IBC, the model presented here can provide a useful testbed to discover biomarkers. Since DCIS is non-invasive, the usage of IBC cells may not be ideal for mimicking DCIS fully. However, since molecular subtypes of DCIS are reported in those of IDC, [12, 47] subtype-specific invasive capacity may be characterized using cancer cells. Recent interests to develop *in vitro* tumor models using microfluidics and tissue engineering have led to many innovative platforms of several cancer types. [22, 23] The usefulness of these systems to precision medicine can be significantly improved with the thorough molecular characterization of cancer cells.

Two TNBC cell lines studied showed increased CD24 expression in IDC-on-chip platforms. CD24 was significantly higher in high-grade DCIS and IDCs than in the non-tumorous breast cells. It was also positively correlated with tumor grade of IDCs. Our results suggest that higher CD24 expression may be associated with malignant transformation and progression in breast cancer biology. Furthermore, higher membranous expression and, in particular, cytoplasmic staining seem to predict malignant transformation, and different patterns of CD24 expression may be associated with different pathological features in breast tumors. [48] In invasive breast carcinomas, CD24 expression was associated with shortened patient overall survival and disease-free survival. [49]

Although the invasion assay using IDC-on chip demonstrate the subtype-specific invasion potential in a context of systematic invasion process involved and controllable manner, it is still limited to recapitulate tumor microenvironment with lack of multiple stroma components fully. Throughout the invasion process, distinct biophysical and biochemical features in the tumor microenvironment (TME) such as hypoxia, the existence of fibroblast, various growth factors, and cytokines play critical roles in regulating the cell response. Specifically, the hypoxia has been reported to enhance cancer cell invasion through hypoxia-induced factor (HIF) activities, which regulate the transcription factors such as Snail, Twist, and matrix metalloproteinase. [50–52] Hypoxia induced Notch signaling was reported to mediate epithelial-mesenchymal transition in breast cancer through enhanced expression of Slug and Snail with e-cadherin suppression. [50] With a benefit of the IDC-on-chip platform to recapitulate the physiological architecture of cancer invasion, the platform can be further improved by applying the complex stroma conditions.

## Supporting information

**S1 Video. A video clip is provided showing local invasion of SUM-159PT into the collagen matrix.**
(AVI)

**S1 File. Invasion characteristics of MDA-MB-231 in IDC-on-chip.** Confocal micrograph of MDA-MB-231 at day 3, cells show membrane protrusions from individual cells as noted with white arrows. Nuclei (blue) and plasma membrane (magenta).
(DOCX)

## Acknowledgments

The authors acknowledge the use of confocal microscopy of Purdue Imaging Facility, Bindley Bioscience Center, Purdue University.

## Author Contributions

**Conceptualization:** Hye-ran Moon, Bumsoo Han.

**Formal analysis:** Hye-ran Moon, Victoria Noe-Kim, Bumsoo Han.

**Funding acquisition:** Bumsoo Han.

**Investigation:** Hye-ran Moon, Natalia Ospina-Muñoz, Victoria Noe-Kim, Yi Yang, Bennett D. Elzey.

**Methodology:** Yi Yang, Bennett D. Elzey, Stephen F. Konieczny.

**Supervision:** Stephen F. Konieczny, Bumsoo Han.

**Visualization:** Hye-ran Moon, Natalia Ospina-Muñoz, Victoria Noe-Kim.

**Writing – original draft:** Hye-ran Moon, Natalia Ospina-Muñoz, Victoria Noe-Kim.

**Writing – review & editing:** Bennett D. Elzey, Bumsoo Han.

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
