## [Decision Letter · Decision Letter 0]

15 Apr 2020

PONE-D-20-05645

Subtype-Specific Characterization of Breast Cancer Invasion using a Microfluidic Tumor Platform

PLOS ONE

Dear Professor Han,

Thank you for submitting your manuscript to PLOS ONE. After careful consideration, we feel that it has merit but does not fully meet PLOS ONE’s publication criteria as it currently stands. Therefore, we invite you to submit a revised version of the manuscript that addresses the points raised during the review process.

We would appreciate receiving your revised manuscript by May 30 2020 11:59PM. To enhance the reproducibility of your results, we recommend that if applicable you deposit your laboratory protocols in protocols.io, where a protocol can be assigned its own identifier (DOI) such that it can be cited independently in the future. For instructions see: http://journals.plos.org/plosone/s/submission-guidelines#loc-laboratory-protocols

We look forward to receiving your revised manuscript.

Kind regards,

Jonghoon Choi, Ph.D.

Academic Editor

PLOS ONE

Journal Requirements:

2. Please ensure that you refer to Figure 5 in your text as, if accepted, production will need this reference to link the reader to the figure.

Reviewers' comments:

Reviewer's Responses to Questions

**Comments to the Author**

1. Is the manuscript technically sound, and do the data support the conclusions?

Reviewer #1: Yes

Reviewer #2: Yes

2. Has the statistical analysis been performed appropriately and rigorously? 

Reviewer #1: Yes

Reviewer #2: Yes

3. Have the authors made all data underlying the findings in their manuscript fully available?

Reviewer #1: Yes

Reviewer #2: Yes

4. Is the manuscript presented in an intelligible fashion and written in standard English?

Reviewer #1: Yes

Reviewer #2: Yes

5. Review Comments to the Author

Reviewer #1: In this paper the authors report a 3D in vitro model of invasive ductal carcinoma using a microfluidic platform. In particular they demonstrate that invasion characteristics of three different human breast cancer cell lines is subtype-specific. The experiments are well performed and the results appear to support the proposed approach. It would be great if the authors could add some experiments with various oxygen concentrations and discussion on effects of hypoxia on invasion characteristics.

Reviewer #2: This manuscript presents quantitative comparison of invasion process in 3D ductal carcinoma in situ (DCIS) cultures in engineered microfluidic platforms. The authors compared 2D and 3D culture of breast cancer cell lines of MDA-MB-231, SUM-159PT and MCF-7 and proved that their platform could have the invasion prediction potential of breast cancer subtypes. If the platform would realize the closer in-vivo-like microenvironments with co-culture and mechanical stress, then the platform could be a useful tool to characterize and predict invasive potential of breast cancer subtypes or patient-derived cells. The manuscript would seem of considerable interest to those working in cancer cell research in engineered microfluidic platforms. After polished based on the following critiques, this manuscript may be able to be published in PLOS ONE. I would recommend that this paper needs minor revision to be published in PLOS ONE. I recommend the current manuscript should revise to include answers for the questions below:

1. There are several human breast cancer cell lines. Why did authors choose MDA-MB-231, SUM-159PT and MCF-7? In particular, MCF-7 instead of MCF 10? Were the cell lines study in 3D cultures in previous studies? Do the results show the similar to results in the previous work or to in vivo behaviors?

2. The authors evaluated invasion characteristics based on local invasion score. Please describe how to estimate local invasion score and explain why the score is valid to evaluate invasion characteristics.

3. A 3D in vitro microvessel model (Matsunaga et al.) was used in this manuscript. Based on the reviewer’s investigation, applying a 3D in vitro microvessel model to breast cancer research is one of the features of this manuscript (Sung et al. Integr Biol 2011; Truong et al. Sci Rep 2016; Choi et al. Lab Chip 2015). If the authors emphasize the benefits of this model to study DCIS, the manuscript would be stronger than the current format.

4. The breast cell invasion responds to ECM stiffness. If the authors describe collagen matrix with the point of view of stiffness, it will help readers to understand microenvironment conditions that induce invasion.

6. PLOS authors have the option to publish the peer review history of their article (what does this mean?). If published, this will include your full peer review and any attached files.

Reviewer #1: No

Reviewer #2: No

---

## [Author Response · Author response to Decision Letter 0]

6 May 2020

Our response to the reviewers is enclosed at the attached files.

---

## [Editor Report · Decision Letter 1]

18 May 2020

Subtype-Specific Characterization of Breast Cancer Invasion using a Microfluidic Tumor Platform

PONE-D-20-05645R1

Dear Dr. Han,

We are pleased to inform you that your manuscript has been judged scientifically suitable for publication and will be formally accepted for publication once it complies with all outstanding technical requirements.

With kind regards,

Jonghoon Choi, Ph.D.

Academic Editor

PLOS ONE
---

## [Editor Report · Acceptance letter]

3 Jun 2020

PONE-D-20-05645R1 

Subtype-Specific Characterization of Breast Cancer Invasion using a Microfluidic Tumor Platform 

Dear Dr. Han:

I'm pleased to inform you that your manuscript has been deemed suitable for publication in PLOS ONE. Congratulations! Your manuscript is now with our production department. 

Kind regards, 

on behalf of

Prof. Jonghoon Choi 

Academic Editor

PLOS ONE